# Transcriptomic Response of Human Nosocomial Pathogen *Pseudomonas aeruginosa* Biofilms Following Continuous Exposure to Antibiotic-Impregnated Catheters

Kidon Sung [1,*], Dan Li [2], Jungwhan Chon [1,†], Ohgew Kweon [1], Minjae Kim [1], Joshua Xu [2], Miseon Park [1] and Saeed A. Khan [1]

1    Division of Microbiology, National Center for Toxicological Research, U.S. Food and Drug Administration, Jefferson, AR 72079, USA; alvarmar@naver.com (J.C.); oh-gew.kweon@fda.hhs.gov (O.K.); minjaekim45@gmail.com (M.K.); miseon.park@fda.hhs.gov (M.P.); saeed.khan@fda.hhs.gov (S.A.K.)

2    Division of Bioinformatics and Biostatistics, National Center for Toxicological Research, U.S. Food and Drug Administration, Jefferson, AR 72079, USA; dan.li@fda.hhs.gov (D.L.); joshua.xu@fda.hhs.gov (J.X.)

\*    Correspondence: kidon.sung@fda.hhs.gov; Tel.: +1-870-543-7527

†    Current address: Department of Pet Total Care, Division of Nursing and Welfare, Kyung-in Women's University, Incheon 21041, Korea.

**Abstract:** Biofilms are complex surface-attached bacterial communities that serve as a protective survival strategy to adapt to an environment. Bacterial contamination and biofilm formation on implantable medical devices pose a serious threat to human health, and these biofilms have become the most important source of nosocomial infections. Although antimicrobial-impregnated catheters have been employed to prevent bacterial infection, there have been concerns about the potential emergence of antibiotic resistance. To investigate the risk of developing resistance, we performed RNA-sequencing gene expression profiling of *P. aeruginosa* biofilms in response to chronic exposure to clindamycin and rifampicin eluted from antibiotic-coated catheters in a CDC biofilm bioreactor. There were 877 and 178 differentially expressed genes identified in planktonic and biofilm cells after growth for 144 h with control (without antibiotic-impregnation) and clindamycin/rifampicin-impregnated catheters, respectively. The differentially expressed genes were further analyzed by Clusters of Orthologous Groups (COGs) functional classification and Kyoto Encyclopedia of Genes and Genomes (KEGG) pathway analyses. The data are publicly available through the GEO database with accession number GSE153546.

**Dataset:** https://www.ncbi.nlm.nih.gov/sra?term=SRP269357

**Keywords:** RNA-seq; *P. aeruginosa* PAO1; clindamycin/rifampicin-impregnated; catheters

## 1. Summary

Bacterial contamination of implantable medical devices, which can be potentially fatal, may also lead to recurrent infections with significant mortality and morbidity for patients [1,2]. Administration of high concentrations of antibiotics or implant replacement may be used to treat infections related to implants, which often fail due to biofilms [3,4]. Hospital-acquired infections in the United States caused 1.7 million infections, 88,000 deaths, and $4.5 billion in health-care costs, and implant-associated infections account for 50–70%

of all hospital-acquired infections [5]. Nosocomial bacteria can attach to the surfaces of medical devices, producing biofilms that have highly reduced susceptibility to the human host immune system [6]. In addition, bacterial cells within biofilms are 10–1000 times more resistant to various antimicrobial agents [7]. Limited penetration of antibiotics into the biofilm and the presence of both persister cells and viable but nonculturable cells are considered to contribute to biofilm-associated antimicrobial resistance [8]. Antimicrobial susceptibility of bacterial biofilms depends on the type of antibiotic, the bacterial strain, and the age of the biofilm [9]. Factors influencing biofilm formation include surface characteristics, such as material type and topography, and host factors, such as antibodies, collagen, fibrinogen, fibronectin, mucin, plasma, and platelets [10,11].

*Pseudomonas aeruginosa* is responsible for more than 2 million hospital-acquired infections and about 90,000 deaths each year [12]. Its biofilms are often found in indwelling medical devices, such as mechanical heart valves, central venous catheters, and Foley catheters, as well as in the lungs, surgical sites, and diabetic, burn, and chronic wounds [12,13]. Exopolysaccharides, extracellular DNA, and cell surface appendages, such as fimbriae, pili, and flagella, play critical functions in biofilm formation by *P. aeruginosa* [14].

A potential way to fight bacterial infections is the development of antimicrobial biomaterials which can thwart biofilm formation. Antimicrobial coatings can prevent bacterial attachment and kill bacteria upon contact with the device surfaces. When they release antibiotics in the surrounding medium, the released antibiotics are delivered to the attached bacterial cells [15]. Commercially available catheters coated with clindamycin, minocycline, or rifampicin have been approved by the US Food and Drug Administration [16]. Previous reports of the survival of *P. aeruginosa* in antibiotic-coated catheters have raised concerns about the potential for developing antimicrobial resistance [17]. Therefore, it is important to understand how antibiotics released from the catheters may affect the survival and antimicrobial resistance of *P. aeruginosa*. Here, we performed total RNA-sequencing of *P. aeruginosa* planktonic and biofilm cells to investigate differential gene expression after exposure to clindamycin/rifampicin-impregnated catheters. Our datasets will elucidate how clindamycin or rifampicin within the catheters affects gene expression associated with antimicrobial resistance of *P. aeruginosa* and which genes are involved in developing antimicrobial resistance.

## 2. Data Description

The RNA sequence-based transcriptomes of *P. aeruginosa* strain PAO1 cultured with uncoated (control) or clindamycin/rifampicin-coated catheters in a CDC biofilm bioreactor were compared. The complete datasets were deposited in NCBI's gene expression omnibus (GEO), accessible through GEO series accession number GEO: GSE153546. The list of all differentially expressed genes can be found in Supplementary Tables S1 and S2. Information about RNA samples of *P. aeruginosa* strain PAO1 cultured with uncoated and antibiotic-impregnated catheters is presented in Table 1. Four biological replicates were used for RNA sequencing.

After removal of duplicate sequences, adapter sequences, and low-quality reads, an average number of 5,205,190 sequence reads were generated. The coverage was about 54X, which was enough for further differential expression analysis. Table 2 exhibits the number of reads after duplicate removal and mapped reads obtained from Bowtie 2 aligner. Figures 1 and 2 illustrate data on the Clusters of Orthologous Groups (COGs) functional classification of the planktonic and biofilm cells grown with control and antibiotic-impregnated catheters. Figures 3 and 4 present top 20 Kyoto Encyclopedia of Genes and Genomes (KEGG) pathways of differentially expressed genes from the planktonic cells grown with control and antibiotic-impregnated catheters.

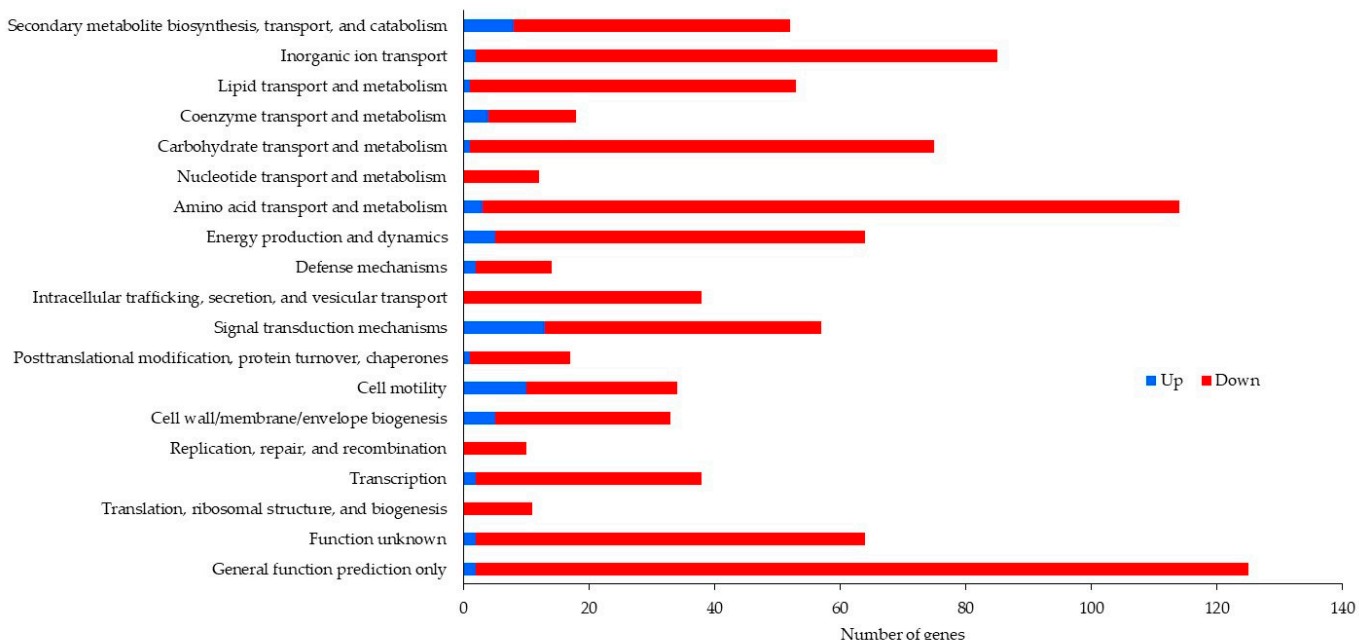

**Figure 1.** COG functional classification of differentially expressed genes from planktonic cells grown with control and antibiotic-impregnated catheters.

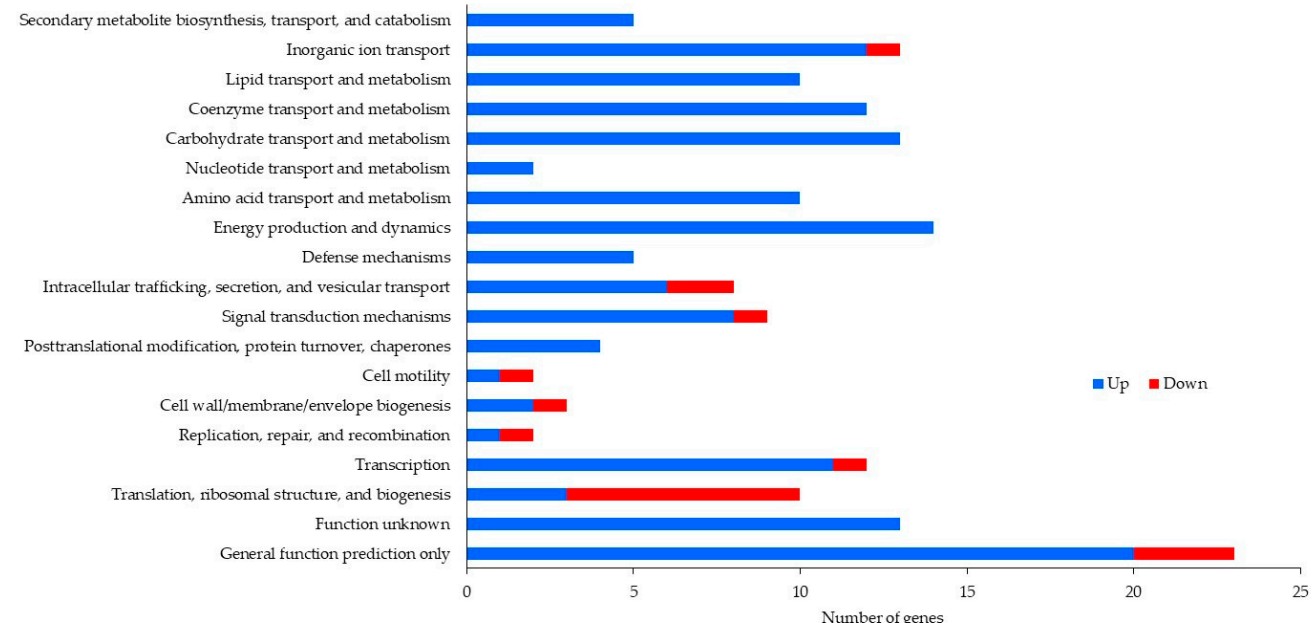

**Figure 2.** COG functional classification of differentially expressed genes from biofilm cells grown with control and antibiotic-impregnated catheters.

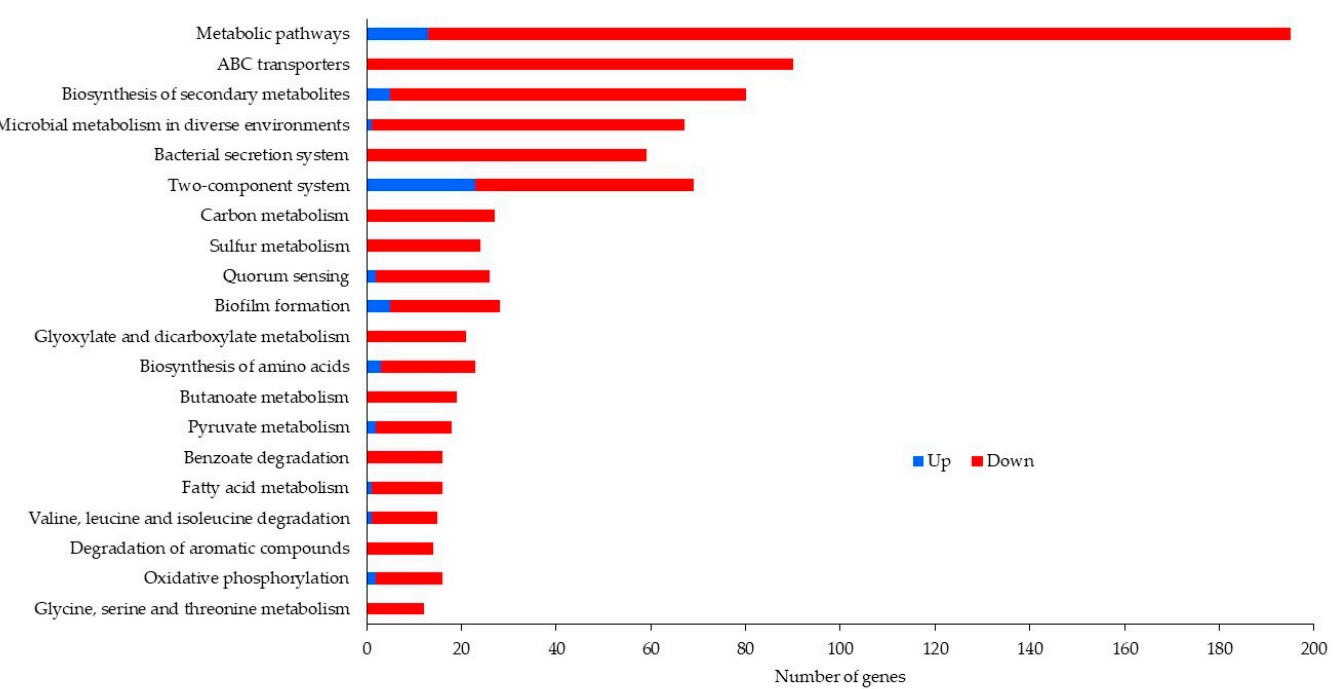

**Figure 3.** Top 20 KEGG pathways of differentially expressed genes from planktonic cells grown with control and antibiotic-impregnated catheters.

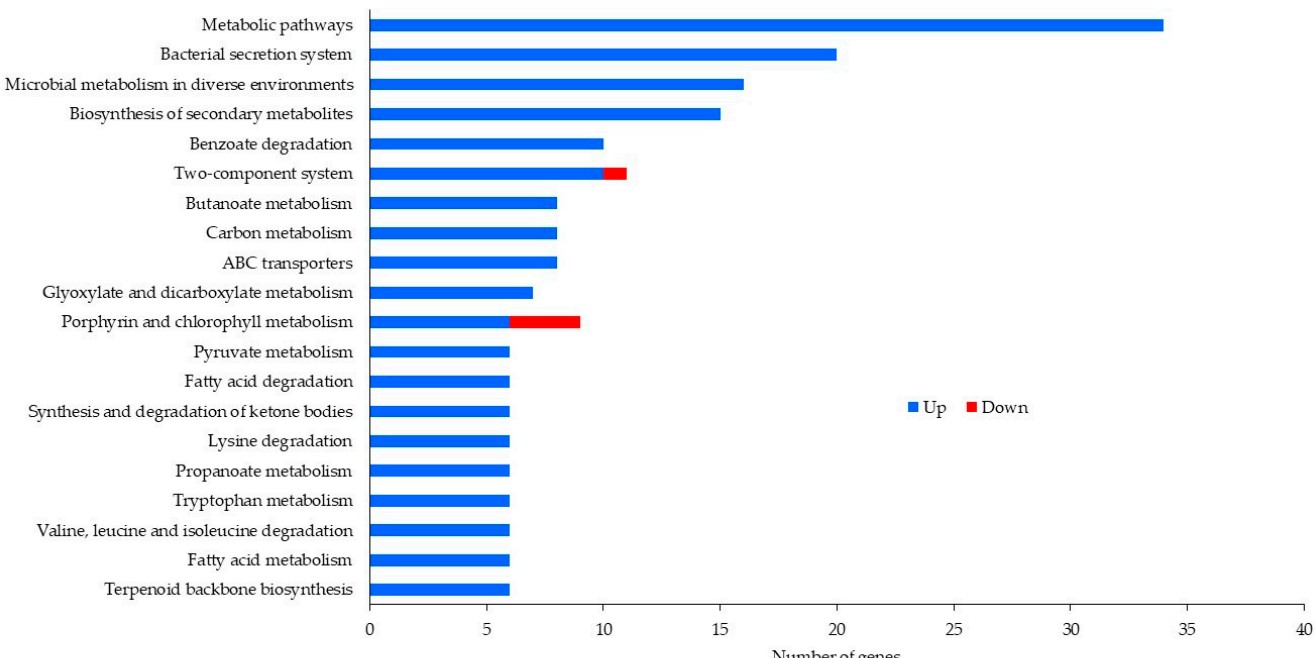

**Figure 4.** Top 20 KEGG pathways of differentially expressed genes from biofilm cells grown with control and antibiotic-impregnated catheters.

**Table 1.** Samples of *Pseudomonas aeruginosa* strain PAO1 cultures grown with control and antibiotic-impregnated catheters.

| Control/Treated. | Growth Mode | RNA Sample Replicates | Accession # |
|---|---|---|---|
| Control catheter | Planktonic cells | CP1 | GSM4646931 |
| | | CP2 | GSM4646932 |
| | | CP3 | GSM4646933 |
| | | CP4 | GSM4646934 |
| | Biofilm cells | CB1 | GSM4646935 |
| | | CB2 | GSM4646936 |
| | | CB3 | GSM4646937 |
| | | CB4 | GSM4646938 |
| Antibiotic-coated catheter | Planktonic cells | AP1 | GSM4646939 |
| | | AP2 | GSM4646940 |
| | | AP3 | GSM4646941 |
| | | AP4 | GSM4646942 |
| | Biofilm cells | AB1 | GSM4646943 |
| | | AB2 | GSM4646944 |
| | | AB3 | GSM4646945 |
| | | AB4 | GSM4646946 |

**Table 2.** Number of reads after duplicate removal and number of mapped reads.

| Library | Number of Mapped Reads | Number of Reads after Duplicate Removal |
|---|---|---|
| CP1 | 30,111,539 | 4,033,623 |
| CP2 | 37,565,923 | 3,451,443 |
| CP3 | 48,299,701 | 6,550,908 |
| CP4 | 53,496,432 | 6,662,813 |
| CB1 | 25,066,031 | 4,731,786 |
| CB2 | 27,158,962 | 5,132,665 |
| CB3 | 20,409,573 | 3,072,031 |
| CB4 | 19,219,978 | 3,096,342 |
| AP1 | 23,962,249 | 6,075,728 |
| AP2 | 22,378,382 | 6,154,300 |
| AP3 | 22,306,149 | 5,329,177 |
| AP4 | 27,637,796 | 7,980,514 |
| AB1 | 29,048,730 | 4,237,760 |
| AB2 | 57,353,487 | 8,311,892 |
| AB3 | 27,382,841 | 3,640,580 |
| AB4 | 31,082,522 | 4,821,482 |

## 3. Methods

### 3.1. Continuous Culture in a CDC Biofilm Bioreactor

About $10^7$ CFU/mL of *P. aeruginosa* strain PAO1 was inoculated in a CDC biofilm bioreactor (BioSurface Technologies, Bozeman, MT, USA) containing catheters without antibiotic coating or with clindamycin/rifampicin coating in tryptic soy broth (TSB) (Thermo Fisher Scientific, Waltham, MA, USA). The culture was mixed at 150 rpm with fresh medium continuously flowing through the bioreactor at a flow rate of 15 mL/min. After incubation for 144 h at 37 °C, the medium was removed and quadruplicate samples of both planktonic and biofilm cells from control and treated cultures were collected.

### 3.2. Total RNA Extraction

Total RNA was extracted using a RNeasy Mini Kit (Qiagen, Germantown, MD, USA) according to the manufacturer's procedure. RNA integrity and quantity were measured using an Agilent 2100 Bioanalyzer (Agilent Technologies, Santa Clara, CA, USA) and a

Qubit Fluorometer (Thermo Fisher Scientific). RNA samples with a Ribosomal Integrity Number (RIN) value greater than 9.0 were considered acceptable in RNA sequencing.

### 3.3. Library Construction, Illumina Sequencing, and Quality Control of Raw Reads

RNA sequencing libraries were prepared with a ScriptSeq Complete Kit from Illumina (San Diego, CA, USA) and validated with the Agilent Bioanalyzer 2100. The library from each sample was sequenced on an Illumina NextSeq 550 sequencer with SE-75. FastQC [18] and Trimmomatic (v0.39) [19] were used for sequencing quality assessment and removal of bad reads (leading and tailing thresholds: 32, minimum length remains: 50 bps). The remaining high-quality reads were fed to Bowtie2 [20] for mapping against *P. aeruginosa* PAO1 reference genome NC_002516.2 released by NCBI. Default parameters of Bowtie2 were applied. Picard was then used to mark and discard the duplicated reads from the mapped data. On average, 17.5% of the total reads remained, which was about 54X of the *P. aeruginosa* genome, after duplicate reads had been removed. The remaining reads were used for differential expression analysis.

### 3.4. Data Analysis

Raw read counts of the annotated genes were calculated using HTSeq-count [21]. All unexpressed genes (median read counts < 10) were excluded from further differential expression analysis. The abundance of genes in each sample was measured as transcript per million (TPM) by Kallisto [22]. To normalize gene expression across samples, we first calculated normalization factors utilizing the expression of housekeeping gene *cheZ* (PA1457) based on the assumption that housekeeping genes were expressed consistently across different samples. The normalization factors of each sample were then used to rescale the expression levels of all of the genes. By using edgeR [23], the log2 fold change was calculated between the treated and control samples and the *p*-value and false discovery rate (FDR) were measured with Student's *t*-test. COG was employed to classify the function of proteins and the KEGG database was used to map the potential metabolic pathway [24,25].

**Supplementary Materials:** The following supporting information can be downloaded at: https://www.mdpi.com/article/10.3390/data7030035/s1. Table S1: Differentially expressed genes of planktonic cells grown with control (CP) and antibiotic-coated (AP) catheters; Table S2: Differentially expressed genes of biofilm cells grown with control (CB) and antibiotic-coated (AB) catheters.

**Author Contributions:** Conceptualization: K.S.; formal analysis: K.S., D.L., J.C., O.K., M.K., J.X. and M.P.; funding acquisition: K.S. and S.A.K.; investigation: K.S., D.L., J.C. and M.P.; supervision: K.S.; writing—original draft: K.S.; writing—review and revise: D.L., J.C., O.K., M.K., J.X., M.P. and S.A.K. All authors have read and agreed to the published version of the manuscript.

**Funding:** This project was supported by the National Center for Toxicological Research and the U.S. Food and Drug Administration (E0759901).

**Acknowledgments:** We recognize Drs. John Sutherland, Jing Han and Jinshan Jin for their critical review of the manuscript and Joanne Berger, FDA Library, for manuscript editing assistance.

**Conflicts of Interest:** The views expressed herein do not necessarily reflect those of the US Food and Drug Administration or the US Department of Health and Human Services. Any mentions of commercial products are for clarification and are not intended as an endorsement.

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
