# Peer review of "Transcriptomic Response of Human Nosocomial Pathogen Pseudomonas aeruginosa Biofilms Following Continuous Exposure to Antibiotic-Impregnated Catheters"

_data, 2022_

Round 1

Reviewer 1 Report

The manuscript provides the data description, the QC and trimming as well as differential expression analysis in sufficient detail. The functional classification as well as KEGG pathway analysis of the differentially expressed genes for planktonic vs biofilm cells also allows the user to visualize the differences and provides a good dataset to study AMR developed by P. aeruginosa in antibiotic-coated catheters. A few minor suggestions:

1) Could you provide the source for the P. aeruginosa PAO1 isolate? That is not mentioned neither in the manuscript or GEO.

2) In Table 1, can you switch the order of the "number of mapped reads" and "number of reads after duplicate removal" columns to reflect in the order of the QC steps?

3) It would be nice (though not a requirement) to include all your QC/trimming and data analysis steps in a bash script and make it available for users of the data through github for easy replication of the steps.

Thanks and good luck!

Author Response

Dear Reviewer:

Comments and Suggestions for Authors

The manuscript provides the data description, the QC and trimming as well as differential expression analysis in sufficient detail. The functional classification as well as KEGG pathway analysis of the differentially expressed genes for planktonic vs biofilm cells also allows the user to visualize the differences and provides a good dataset to study AMR developed by P. aeruginosa in antibiotic-coated catheters. A few minor suggestions:

Authors highly appreciate the reviewer’s thorough reading of the manuscript and helpful remarks that helped us to improve the manuscript. We have seriously considered and addressed your valuable comments by responses in the following context. We hope our revision has improved the paper to a level of your satisfaction.

1) Could you provide the source for the P. aeruginosa PAO1 isolate? That is not mentioned neither in the manuscript or GEO.

 Response: Pseudomonas aeruginosa PAO1 had been isolated in 1954 from a wound in Melbourne, Australia (Holloway, B. W. Genetic recombination in Pseudomonas aeruginosa. J. Gen. Microbiol. 1955, 13:572-581). This PAO1 strain has become the most widely used reference strain for Pseudomonas genetics and functional analyses because physical and genetic maps were available (Schmidt, K. D., Tümmler, B. & Römling, U. Comparative genome mapping of Pseudomonas aeruginosa PAO with P. aeruginosa C, which belongs to a major clone in cystic fibrosis patients and aquatic habitats. J. Bacteriol. 1996, 178:85–93).

2) In Table 1, can you switch the order of the "number of mapped reads" and "number of reads after duplicate removal" columns to reflect in the order of the QC steps?

Response: The order of the "number of mapped reads" and "number of reads after duplicate removal" columns was switched as the reviewer advised.

3) It would be nice (though not a requirement) to include all your QC/trimming and data analysis steps in a bash script and make it available for users of the data through github for easy replication of the steps.

Response: We tried to put QC/trimming and data analysis steps in GitHub but our bioinformatic team doesn’t know how to do it. So, we made it a supplementary file for use by any interested researcher.

Reviewer 2 Report

Dear Authors,

The research is conducted very well and written in a clear context. It seems useful and all datasets are complete, but it is still difficult to understand the impact of this relevant data for further research. Maybe you can comment to me, what to do with this high-quality data for further research in a few sentences in the summary? 

Kind regards,

Reviewer

Author Response

Dear Reviewer:

Comments and Suggestions for Authors

The research is conducted very well and written in a clear context. It seems useful and all datasets are complete, but it is still difficult to understand the impact of this relevant data for further research. Maybe you can comment to me, what to do with this high-quality data for further research in a few sentences in the summary?

Response: Authors highly appreciate the reviewer’s thorough reading of the manuscript and helpful remarks that helped us to improve the manuscript. Two commercially available catheters impregnated with combinations of clindamycin/rifampicin and minocycline/rifampicin have been approved by the US Food and Drug Administration (FDA). It has been reported that clindamycin/rifampicin- and minocycline/rifampicin-coated catheters had no effect on the growth of P. aeruginosa. These results have increased concern over the possible contamination of P. aeruginosa and development of antimicrobial resistance to the antibiotics impregnated in the catheters. Therefore, it is important to understand whether the presence of the antimicrobial in the medical devices can lead to antibacterial resistance. Our datasets will elucidate how clindamycin or rifampicin within the catheters affects gene expression associated with antimicrobial resistance of P. aeruginosa and which genes are involved in developing antimicrobial resistance.

The following sentences was added to the summary. “Our datasets will elucidate how clindamycin or rifampicin within the catheters affects gene expression associated with antimicrobial resistance of P. aeruginosa and which genes are involved in developing antimicrobial resistance.”
